# Using the C-Read as a Portable Device to Evaluate Reading Ability in Young Chinese Adults: An Observational Study

**DOI:** 10.3390/jpm13030463

**Published:** 2023-03-01

**Authors:** Tian Cheng, Taikang Yao, Boxuan Xu, Wanwei Dai, Xuejiao Qin, Juan Ye, Lingge Suo, Chun Zhang

**Affiliations:** 1Peking University Health Science Center, Peking University, Beijing 100191, China; 2Department of Ophthalmology, Peking University Third Hospital, Beijing 100191, China; 3Beijing Key Laboratory of Restoration of Damaged Ocular Nerve, Peking University Third Hospital, Beijing 100191, China; 4Department of Ophthalmology, The Second Hospital of Shandong University, Jinan 250033, China; 5Department of Ophthalmology, The Second Affiliated Hospital Zhejiang University, Hangzhou 310058, China

**Keywords:** C-Read, reading speed, smartphone-based application, healthy subjects

## Abstract

We evaluated the reading characteristics of normal-sighted young adults using C-Read to provide baseline healthy population values. We also investigated the relationship between the National Eye Institute’s Visual Functioning Questionnaire (VFQ-25) score and reading ability, myopia, and hours of screen use, focusing on the extent to which these factors affect participants’ visual function and, ultimately, their vision-related quality of life (QoL). Overall, 207 young, healthy participants (414 eyes) aged 18–35 years were tested for reading speed using C-Read connected to a smartphone-based application between December 2022 and January 2023. Each participant received a VFQ-25 questionnaire to evaluate vision-related QoL. Data on daily e-screen usage hours were collected. Among the participants, 91 (44.0%) were women; their mean (SD) age was 22.45 (4.01) years. The mean (SD) reading acuity (RA) was 0.242 (0.124), 0.249 (0.120), and 0.193 (0.104) logarithmic minimum angle of resolution (logMAR) for the right, left, and both eyes, respectively. The mean (SD) maximum reading speed (MRS) was 171.65 (46.27), 168.59 (45.68), and 185.16 (44.93) words per minute (wpm) with the right, left, and both eyes, respectively. The mean (SD) critical print size (CPS) was 0.412 (0.647), 0.371 (0.229), and 0.419 (1.05) logMAR per the right, left, and both eyes, respectively. The RA and CPS were significantly different between sexes (*p* = 0.002 and *p* = 0.001). MRS was significantly different between the education level (*p* = 0.005) and myopia level groups (*p* = 0.010); however, it was not clear whether this difference was confounded by age. The myopic power in diopters significantly affected RA (coefficient, −0.012; 95% CI, −0.018 to −0.006; *p* = 0.001); screen time significantly affected MRS (coefficient, 0.019; 95% CI, 0.57 to 6.33; *p* = 0.019). RA (coefficient, −21.41; 95% CI, −33.74 to −9.08; *p* = 0.001) and duration of screen use (coefficient, -0.86; 95% CI, −1.29 to −0.43; *p* < 0.001) independently had a significantly negative correlation with VFQ-25 scores. Our findings provide a baseline value for C-Read in normal-sighted young adults. Refractive status significantly affected RA, while screen time significantly affected MRS. Interventions aimed at enhancing RA may have the potential to maximize vision-related QoL and enable older adults with impaired vision to achieve greater outcomes. Future, larger-scale, C-Read experiments will help provide newer, more optimal methods for the early diagnosis of visual impairment.

## 1. Introduction

Reading is a core function of vision. Currently, the epidemic increase in the prevalence of myopia among school-aged children and the increase in the number of older adults with age-related conditions (e.g., presbyopia, glaucoma, age-related macular degeneration) contribute to challenging public health conditions worldwide [1]. It is projected that by 2050 half of the world’s population will be myopic [2]. There is conclusive evidence that visual impairment can lead to reading difficulty and directly affect quality of life (QoL) [3]. However, the relationship between myopia and reading is less adequately studied, although myopia is a risk factor for visual impairment. Kandel et al. [4] pointed out that myopia can cause difficulties in reading, driving, sports, and entertainment activities, which in turn reduce quality of life.

According to a large study of children and adults of all ages, 62% of patients seeking low-vision consultation were primarily interested in improving their reading ability [5]. Reading ability is often used as an outcome indicator in clinical trials to assess the effectiveness of treatments, surgical procedures, and rehabilitation techniques [6,7,8]. In addition, reading ability is easy and inexpensive to assess and can be administered outside the hospital. It is, therefore, an appropriate choice for early self-testing.

Numerous continuous-text reading-ability tests have been developed and applied in vision health care, most notably the Minnesota Low-Vision Reading Test (MNREAD) [9,10] and Radner Reading charts [11,12]. Both tests quantify reading ability by measuring three core reading ability indicators [13]: reading acuity (RA): the smallest print that can just be read; maximum reading speed (MRS): the reading speed when performance is not limited by print size; and critical print size (CPS): the smallest print that supports the maximum reading speed. However, the MNREAD and Radner Reading charts are available only in English, limiting the application of these scales in non-English-speaking countries.

Han et al. [14] developed and validated the Chinese reading-ability test chart, C-Read, and its supporting electronic portable device to assess Chinese reading ability. There are three scales in C-Read, each consisting of 16 12-character simplified Chinese sentences acquired from first- to third-grade textbooks, and they have been tested for homogeneity [14]. The C-Read derives some time-tested design principles from the aforementioned tests, including the standardized continuous text that closely resembles daily reading materials, high-frequency vocabulary at the primary reading level, the most streamlined typefaces, logarithmic progression of print sizes, and consistency in the spatial arrangement of sentences [10,15]. In addition, given the many differences between logographic Chinese and linear alphabetic Latin languages, such as sentence composition, the number of syllables, and the use of Chinese characters with simple and complex strokes, C-Read adapted the above design principles based on an in-depth understanding of these features of Chinese text reading [14]. The reliability of the C-Read has also been tested using passages from the International Reading Speed Text (IReST) [16]; the results showed that C-Read scores could accurately predict passage reading ability over a wide range of refractions [17]. Hence, this indicated that C-Read is a reliable and valid clinical tool for quantitatively assessing the reading ability of readers of simplified Chinese characters.

Another major advantage of C-Read is that it allows continuous-text reading ability tests on an electronic screen. This advantage facilitates data processing and meets the increasing reality of digital reading for citizens in the 21st century [18]. Digital versions of reading tests are also advantageous because text size, contrast polarity, color, and font size can be easily adjusted. However, the performance of any digital reading test largely depends on screen size, display technology, and resolution [19]. MNREAD and Radner reading tests have been shown to have minimal differences between the evaluation parameters of the digital and traditional versions [2,20,21,22]. Therefore, it is dependable to use electronic devices for reading-ability tests.

Little is known about the reading characteristics of young adults with normal vision whose native language is Chinese, due to the lack of a reliable Chinese reading chart to assess Chinese reading ability. Han et al. [14] measured the reading characteristics of 118 Chinese college students between the ages of 18–24 who had normal or corrected vision and obtained normal mean (SD) reference values for RA, CPS, and MRS of 0.16 (0.05) logMAR, 0.24 (0.06) logMAR, and 273.44 (34.37) Chinese wpm, respectively. In addition, previous studies have observed a clear correlation between reading speed and vision-related QoL in people with eye disease, and the results of a study by an Italian academic [23] point to both reading speed and visual acuity as important determinants of quality of life. However, it remains to be seen whether this correlation holds in the healthy population. The objectives of this study were (1) to evaluate the reading characteristics of normally sighted young adults over a wider age range (18–35 years) using a C-Read test to provide baseline values in young Chinese adults, and (2) to investigate the relationship between the VFQ-25 scale and reading ability, myopia, and screen-use hours in young Chinese adults, looking at how these factors affect visual function and, ultimately, vision-related QoL.

## 2. Materials and Methods

### 2.1. Study Design

This single-center observational study was approved by the Institutional Review Board of Peking University Third Hospital (IRB00006761-M2022800) and followed the tenets of the Declaration of Helsinki. Informed consent was obtained from all participants after the experimental procedures were fully described. The trial was registered with the Chinese Clinical Trial Registry (NCT5673954) and followed the Consolidated Standards of Reporting Trials (CONSORT) reporting guidelines.

### 2.2. Participants

Figure 1 shows the flow diagram of the participants according to the CONSORT statement [24]. We set σ = 7δ and α = 0.05, ran a two-sided test, and worked out that the sample needed 189 people. Considering a 5% missing rate, a total of at least 200 people needed to be recruited at first. In total, 219 healthy participants were recruited for this study at Peking University between December 2022 and January 2023. To reduce selection bias due to non-random willingness to participate in the survey, participants were recruited by simple random sampling from different academic departments at Peking University (including Humanities, Science, Social Sciences, and Medicine). Healthy participants were included according to the following criteria: (1) age between 18 and 35 years; (2) best-corrected visual acuity (BCVA) ≥1.0 decimal by using standard logarithmic visual acuity chart; (3) good physiological functioning of the body’s systems without health problems requiring medical intervention; (4) no strabismus, amblyopia, or any other ocular or systemic diseases that may affect reading ability; (5) good cooperation in the questionnaire survey and reading tests. Professional ophthalmologists examined the subjects for distance and near vision via slit lamp observation and indirect ophthalmoscopy. All subjects’ eye health status was confirmed by professional ophthalmologists.

### 2.3. Screen Time and Visual QoL

A questionnaire was designed to collect the participants’ sex, age, refractive status, and time spent on screens, including personal computers, phones, and pads (see Appendix A). Furthermore, the VFQ-25 [25] was chosen to collect and assess visual QoL. The VFQ-25 is a reliable questionnaire used to evaluate vision-related QoL globally [26,27,28,29,30]. It has 12 subscales that cover general health, general vision, eye pain, near activities, distance activities, social functioning, mental health, role difficulties, dependency, driving, color vision, and peripheral vision. In this study, the Chinese version of the VFQ-25 developed by Chan et al., was used [30].

### 2.4. C-Read Device and Smartphone-Based App

The procedure is illustrated in Figure 2. The C-Read test was performed in a well-lit room with the device placed on a special stand that was vertically fixed at eye level. The mean screen luminance of the C-Read device is 150–250 cd/m^2^. The participants sat 40 cm from the screen, with their heads on the headrest, verbally reading sentences from the screen’s top row (largest print size). They were instructed to read these sentences as fast as possible without making any errors and encouraged to read as many characters as possible when encountering sentences with smaller print sizes. In addition, a triangular guide symbol appeared on the screen at the beginning of each sentence indicating their location. The test was stopped when the participants reported that they could no longer read any character.

The C-Read application on the smartphone recorded the time each participant spent reading the sentences aloud and the number of correct Chinese characters read. The C-Read application on the phone was connected to the C-Read device using Bluetooth. Three scales, A, B, and C, were used to assess participants’ binocular, right-eye, and left-eye reading ability. We used scale A to test both eyes; scale B to test the left eye; scale C to test the right eye. When changing eyes, the examiner could switch to a different scale for the next test using the C-Read application on the smartphone (see Appendix A).

The measuring principle of C-Read was as follows: the reading speed vs. print size curves obtained in the C-Read test for normal and low-vision subjects have the same typical shape [9]. Over a wide range of print sizes, the reading speed remains constant, forming a platform representative of MRS. When the print size reaches a certain value and the reading speed decreases rapidly, the print size is called CPS. The minimum print size that can be read is defined as the RA. C-Read used a hybrid algorithm to obtain the reading characteristics, that is, using a bilinear fitting algorithm to determine the critical character size and maximum reading speed and using the MNREAD method to calculate RA [14].

The smartphone uploaded all the experimental data (the number of Chinese characters correctly read and the time spent reading them) under different print sizes, recorded in the C-Read application program, to the cloud storage and calculated the corresponding reading speeds under different print sizes. These raw data were used to plot the print size versus reading speed graph, and the program automatically calculated the RA, MRS, and CRS values. The participants’ reading abilities could be assessed by analyzing these reading characteristics.

### 2.5. Data Collection and Statistical Analysis

All questionnaires were collected after obtaining informed consent from the subjects. The researcher briefed the participants on the purpose and methods of the experiment. Most of the information was collected as completed online questionnaires, and some of it was collected using paper questionnaires. Data from all questionnaires carefully completed by the participants were included in the analysis.

Data analysis was performed using the STATA Statistics Software (Version 17.0; Stata Corp., College Station, Texas, USA). Figures were drawn using GraphPad Prism (version 7.0; GraphPad Software, Inc., San Diego, CA, USA). The distribution of the variables was judged using Kolmogorov–Smirnov test. Mann–Whitney test and Kruskal–Wallis test were used to analyze differences in reading ability between gender, education, and myopia levels. Factors affecting reading ability and the relationship between reading ability and QoL were analyzed using multiple linear regression models and stepwise adjustment. Statistical significance was set at *p* < 0.05.

## 3. Results

### 3.1. Demographics

In total, 219 healthy participants were surveyed. Finally, 207 participants returned the completed questionnaire, resulting in a response rate of 94.5% (207/219). The age range of the participants who met the eligibility criteria was 18–35 (median, 21 years; interquartile range, 19–24 years).

Among the 207 participants, 91 (44.0%) were female, and the mean (SD) age was 22.45 (4.01). Participant characteristics are listed in Table 1. Myopia between −0.00 and −0.50 diopters is classified as emmetropia. Low, moderate, and high myopia describe myopia between −0.50 and −3.00, −3.00 and −6.00, −6.00 or more diopters, respectively [31]. All participants were native Chinese speakers. None of the studies had gender restrictions on sampling; therefore, the unequal number of males and females reflects the nature of the volunteers.

### 3.2. Characteristics of Reading Speed

Table 2 shows the participants’ reading characteristics. The mean (SD) RA for the young, healthy population was 0.242 (0.124), 0.249 (0.120), and 0.193 (0.104) logMAR in the right, left, and both eyes, respectively. The mean (SD) MRS for the young, healthy population was 171.65 (46.27), 168.59 (45.68), and 185.16 (44.93) wpm in the right, left, and both eyes, respectively. The mean (SD) CPS for the young, healthy population was 0.412 (0.647), 0.371 (0.229), and 0.419 (1.05) logMAR in the right, left, and both eyes, respectively. Figure 3 shows the reading characteristics for both, left, and right eyes for different sexes, education levels, and myopia levels. We found that RA and CPS significantly differed between the sex groups (Figure 3a,c; *p* = 0.005, *p* = 0.025). CPS showed a significant difference among the myopia groups (Table 3; *p* < 0.001) in a single eye and tended to increase with myopia progression (Figure 3i). In addition, MRS was statistically different between the education groups (Table 3; *p* = 0.002), but it was unclear whether this difference was confounded by age. We also analyzed the data from the left and right eyes mixed together as a single eye and obtained exactly the same results (see Appendix A).

### 3.3. Effect of Age, Myopia, and Screen Time on Reading Ability

Table 4 shows the multivariate linear regression model for the correlation between monocular reading characteristics, age, screen time, and monocular myopic refraction in diopters. After adjustment using stepwise regression, we found that the myopic power had a significant effect on RA (Figure 4a; coefficient, −0.012; 95% CI, −0.018 to −0.006; *p* = 0.001), screen time had a significant effect on MRS (Figure 4b; coefficient, 0.019; 95% CI, 0.57 to 6.33; *p* = 0.019), and both myopic refraction (diopters) and screen time had a significant effect on CPS (Figure 4c; diopters: coefficient, 0.015; 95% CI, 0.004 to 0.026; *p*< 0.001; screen time: coefficient, −0.014; 95% CI, −0.021 to −0.006; *p* = 0.010). Parameter estimates for the MLR models of RA, MRS, and CPS for right, left, and both eyes are presented in Appendix A. However, since the participants recruited in this research were all young, healthy people, the insignificance of age does not mean that age does not affect these reading characteristics.

### 3.4. Effect of Reading Ability on VFQ-25 Score Results

Reading is an important part of visual life, and a decline in the reading function directly impacts a patient’s QoL. While clinicians are primarily concerned with changes in pathological factors, patients are interested in how these factors affect their functional status and vision-related QoL. The VFQ-25 is used to measure the vision-specific quality of life, such as stereoacuity, light adaptation, and dark adaptation, and to evaluate the overall quality of survival states, including self-care, activity, and social and psychological parameters.

Figure 5 shows the regressions of VFQ-25 scores using data from the right, left, and both eyes. The independent variables included age; visual acuity; the amount of time spent on computers, phones, and pads; and three criteria of reading ability. Furthermore, using a stepwise adjustment process, two significant independent variables were selected: RA (Table 5; coefficient, −21.41; 95% CI, −33.74 to −9.08; *p* = 0.001) and the duration of screen use (Table 5; coefficient, −0.86; 95% CI, −1.29 to −0.43; *p* < 0.001). Both had negative correlation coefficients with VFQ-25 scores, which could be explained by the fact that longer screen time would lead to a greater impact on vision-related functions. In addition, a larger RA value would indicate a greater decline in visual acuity, which could also contribute to a lower VFQ-25 score.

## 4. Discussion

Reading performance predicts visual ability and vision-related QoL [32]. Although there is no consensus on the best way to assess reading performance, most tests have identified a common set of characteristics [32]: (i) MRS and RA are key outcome variables, while tests of comprehension or reading stamina are reserved for specific research questions; (ii) reading speed is measured for meaningful text, although this may allow cognitive factors to have a greater impact; and (iii) to ensure that the text is read accurately, reading aloud rather than silently is preferred. Reading aloud also facilitates scoring [33].

In 1854, Jaeger invented the Near Vision Scale, marking the first clinical reading test [34]. In the century and a half that followed, dozens of reading-ability tests appeared, but there were few Chinese versions of these tests. In addition to the C-Read from MNREAD, the only other tests are the IResT, which was translated into 19 languages, including Chinese by Klosinski et al. [35] in 2012, and the Chinese reading-speed test for children by Cheung et al. [36] in 2015. Few studies of these Chinese-language tests were based on smartphone applications, which measure close-reading ability.

With technology advancement, more tests with multiple functions to assess close-reading ability were used, and the widely accepted ones included MNREAD and IReST. In general, IReST is more advantageous for measuring reading speed for fixed print sizes [35], and the MNREAD Acuity Test is widely used for measuring reading speed for texts of different sizes [32]. The Chinese version of the reading speed scale C-Read was developed on the basis of MNREAD. It used sentences of a standardized reading level and length, acquired the reliability and stability of the MNREAD test, and had good test repeatability, making it a reliable measure of Chinese reading ability [14]. Moreover, through improvements in the medium, C-Read could support testing on higher-resolution e-readers via a smartphone-based application, which is certainly more relevant nowadays as e-reading time is significantly longer.

Notably, a “floor effect” is frequently observed in the C-Read test. Here, participants can only correctly read one or two characters per sentence of the two or three smallest characters before they stop trying, resulting in a floor effect at the small-character-size end of the reading speed graph [37]. This effect may be unique to simplified Chinese reading because of the presence of simple and readable characters in these sentences. To correct the effect caused by the floor effect, the C-Read program automatically removes all characters in the floor effect except for the largest character in size, producing a monotonically rising linear part of the reading speed function used to calculate the C-Read parameter [14].

Reading performance is one of the most important outcome indicators of the effectiveness of therapeutic interventions and vision rehabilitation. Despite the many differences between highly standardized clinical reading tests and daily reading, clinical test scores are good predictors of reading ability [38]. We aimed to assess the performance of university students with normal vision in the C-Read test, providing baseline values for that test in a healthy population. Our reference values for reading characteristics in the healthy young population were lower than those of Han et al. [14]. This may be due to the inclusion of a wider age range in our participants, particularly those over 30 years old. This finding also reinforces the strong association between reading ability and age.

Reading characteristics provide interesting insights into the functional performance of patients before and after therapy. However, we expected them to be used more frequently as an early screening test for vision loss disorders. Most visual function questionnaires ask patients to self-report assessments of their reading difficulties. Although self-reported ability is usually consistent with measured reading performance, there can be discrepancies, especially when patients self-perceive no reading difficulties but test results indicate RA or MRS impairment. This inconsistency between self-perception and examination results may indicate preclinical visual impairment and requires further examination [39].

Another project explored the relationship between the time spent on electronic screens and reading characteristics. We chose college students because the prevalence of myopia among college students is high and they tend to be younger [40,41,42]. Students are also extensive users of electronic devices, and their exposure to screen terminals is increasing and diversifying. Notably, online learning during home isolation after the COVID-19 outbreak has significantly prolonged the electronic screen exposure of college students [17,43]. These students faced a huge visual burden, leading to eye fatigue, blurred vision, eye dryness, or myopia [44,45,46]. However, the relationship between screen-viewing behavior and myopia is unclear or even contradictory [42,47,48,49]. A plausible reason is that the existing research on screen-viewing behavior is not sufficiently accurate and comprehensive, and the evaluation methods have difficulty considering the intensity and duration of screen use.

Our study initially revealed that the myopic power is a key factor affecting RA, presumably because both are physical quantities indicating visual acuity. The negative correlation coefficient between screen time and MRS can be well explained by the fact that long-duration eye use causes eye strain, reducing reading speed. CPS is associated with both myopic power and screen time, which reflected the joint effect of visual acuity and eye strain on the turning point in close reading. In addition, Calabrèse et al. [50] pointed out that age was also related to reading characteristics; however, due to the age limit of this study, we did not find a significant correlation between age and reading characteristics.

RA and CPS showed significant differences between sex groups. The significant differences in RA and CPS between sex groups may reflect differences in eye-tracking habits between women and men. Our study found that women spent an average of 0.73 h (11.78 for women and 11.05 h for men, *p* = 0.040) more time using screens daily than men and that longer screen use may account for the lower average RA and CPS in women than in men (see Appendix A). The differences in MRS scores between the education and myopia arrays were significant. Although it was unclear whether this difference was affected by age, it might reflect, to some extent, the influence of education and learning experience on reading ability. Both RA and CPS increased with myopia level; however, the difference among the various myopia level groups was insignificant, which might be related to the insufficient sample size.

Two variables, RA and the amount of screen time, had negative correlation coefficients with VFQ-25 scores. We collected both participants’ subjective perceived screen time and objective daily personal computer, phone, and pad usage time, finding that the latter negatively affected the visual QoL, whereas the former had no such effect. Chan et al. [51] reported that the visual acuity of the better eye is highly correlated with the quality of life. As mentioned above, longer screen exposure predisposes patients to myopia with decreased RA, whereas deterioration in visual function may result in a lower VFQ-25 score. We also observed that the VFQ-25 score gradually decreased with an increase in myopia level, which is consistent with previous studies (see Appendix A) [52]. There was no significant difference in VFQ-25 scores between the sexes.

Zhu et al. [52] found that educational attainment was also an important demographic factor affecting the VFQ-25 score because patients with higher educational attainment might know more about eye conditions and seek suitable treatment before irreversible damage occurs. It has been reported that knowledge of eye diseases is an important positive predictor of QoL [53]. However, in this study, no significant difference was observed between educational level and VFQ-25 score, which might be because the subjects participating in the study were all well-educated college students.

The limitation of this study is the lack of age-related controls. The subjects in this study were all in the 18–35 age group and there were no data for children or middle-aged or elderly healthy subjects, which may explain why we did not observe a negative correlation between VFQ-25 score and age. Further experiments covering subjects with wider age ranges to explore the effect of age on reading characteristics and visual QoL are necessary. However, our study initially investigated the negative correlation between screen time and RA and VFQ-25 scores, which led us to question whether long-term electronic screen use has an impact on eye health consequences and whether there is a time point beyond which visual health deteriorates rapidly with screen use.

## 5. Conclusions

Our findings provide baseline values for C-Read in healthy individuals. Myopic power was a significant factor affecting RA, while screen-use time significantly affected MRS. CPS was associated with both myopic power and screen time. In addition, RA and CPS significantly differed between sex groups, increasing with the progression of myopia. Finally, MRS scores were significantly different between different education and myopia level groups.

A person’s RA can significantly affect their vision-related QoL, suggesting that interventions aimed at enhancing RA might have the potential to maximize visual quality. This would enable some older adults with impaired vision to achieve better outcomes from low-vision rehabilitation. Total screen time negatively affected vision-related QoL, although participants perceived that screen-use time had no such effect. More research is needed to investigate the relationship between RA, screen time, and vision-related QoL.

Reading performance is an important indicator of reactive visual functioning. A significant difference between a patient’s self-assessed reading ability and measured reading ability may indicate preclinical visual impairment. Future, larger-scale, C-Read experiments will help determine thresholds for normal and abnormal reading speeds and provide newer, more optimal methods for the early diagnosis of visual impairment.

## Figures and Tables

**Figure 1 jpm-13-00463-f001:**
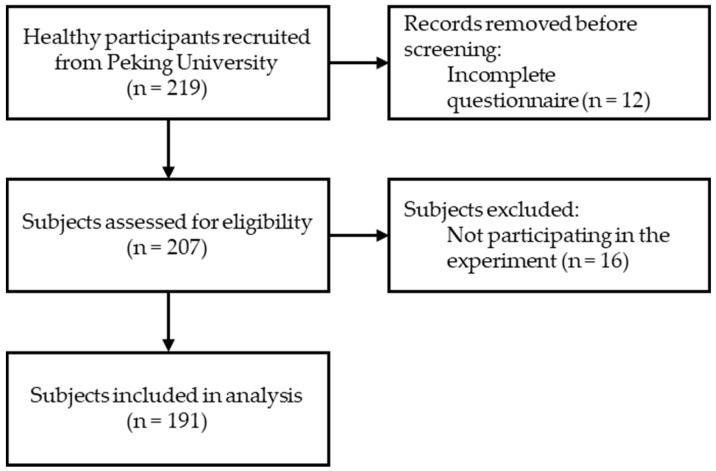
Flow Diagram of Participants According to CONSORT Statement.

**Figure 2 jpm-13-00463-f002:**
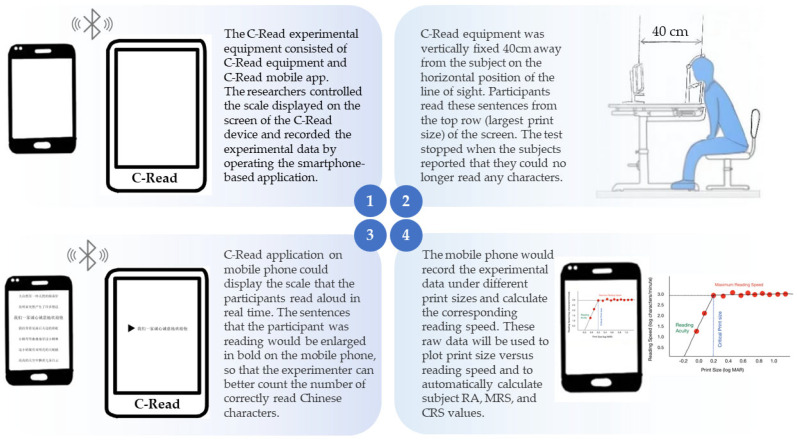
Flow Diagram of Trial Design. Abbreviations: RA, reading acuity; MRS, maximum reading speed; CPS, critical print size.

**Figure 3 jpm-13-00463-f003:**
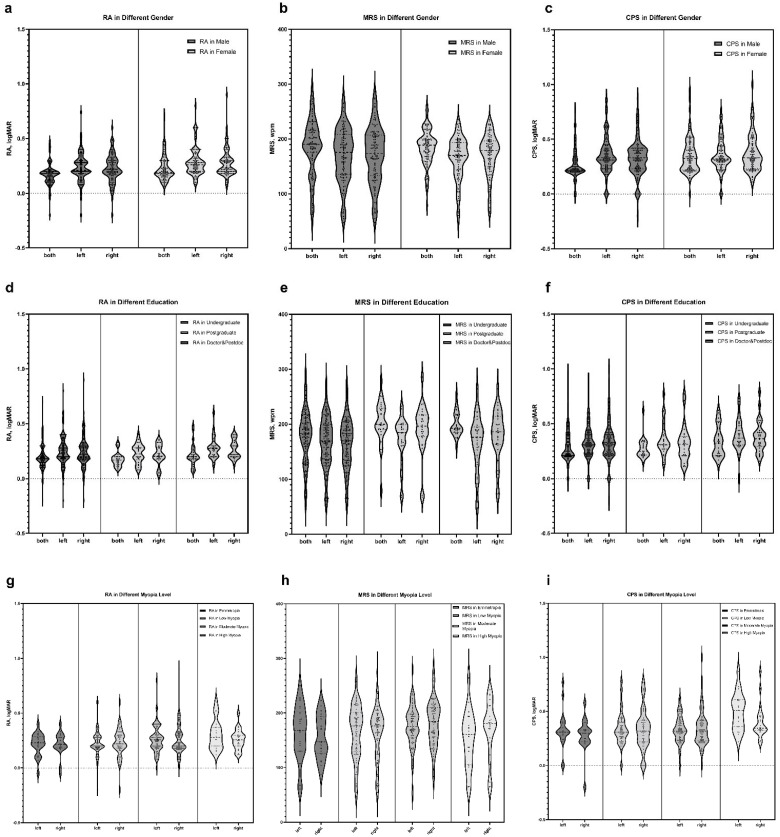
Reading characteristics for both, left, and right eyes for different sexes, education levels, and myopia levels. (**a**) The RA for different sexes was statistically significant (*n* = 191, *p* = 0.005), as determined by the two-tailed Mann–Whitney test. (**b**) The MRS for different sexes was not statistically different (*n* = 191, *p* = 0.685), as determined by the two-tailed Mann–Whitney test. (**c**) The CPS for different sexes was statistically significant (*n* = 191, *p* = 0.025), as determined by the two-tailed Mann–Whitney test. (**d**) The RA of the three education levels was not significantly different (*n* = 189, *p* = 0.217), as determined by the Kruskal–Wallis test. (**e**) The MRS for the three education levels was statistically significant (*n* = 189, *p* = 0.002), as determined by the Kruskal–Wallis test. (**f**) The CPS for different education levels was not statistically different (*n* = 189, *p* = 0.093), as determined by the Kruskal–Wallis test. (**g**) The RA for the different myopia levels was not statistically significant (*n* = 382 eyes, *p* = 0.072), as determined by the Kruskal–Wallis test. (**h**) The MRS for different myopia levels was statistically significant (*n* = 382 eyes, *p* = 0.049), as determined by the Kruskal–Wallis test. (**i**) The CPS for different myopia levels was statistically significant (*n* = 382 eyes, *p* < 0.001), as determined by the Kruskal–Wallis test. Abbreviations: RA, reading acuity; MRS, maximum reading speed; CPS, critical print size.

**Figure 4 jpm-13-00463-f004:**
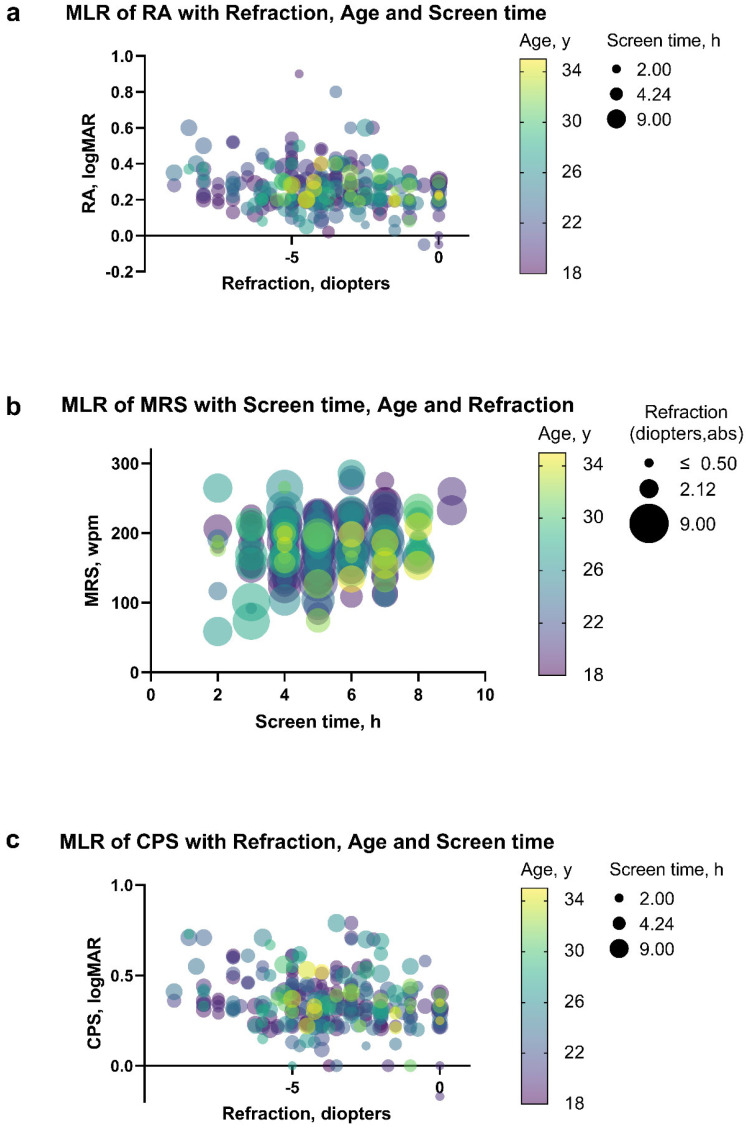
Parameter estimates for multivariate linear regression model for the outcomes RA, MRS, and CPS. (**a**) RA as a function of myopic power, age, and screen time. Points show both eyes’ RA for each participant tested. The age of participants is color-coded, and the screen time is size-coded. (**b**) MRS as a function of screen time, age, and refraction (diopters). Points show both eyes’ MRS for each participant tested. The ages of participants are color-coded, and the myopic power of participants is size coded. (**c**) CPS as a function of refraction (diopters), age, and screen time. Points show both eyes’ CPS for each participant tested. The age of participants is color-coded, and the screen time is size-coded. Model estimates and their 95% CI are given in Table 4. Abbreviations: abs, absolute value; MLR, multiple linear regression; RA, reading acuity; MRS, maximum reading speed; CPS, critical print size.

**Figure 5 jpm-13-00463-f005:**
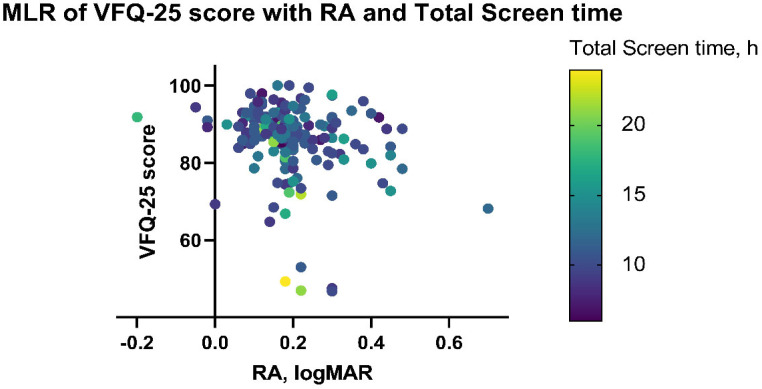
Parameter estimates for multivariate linear regression model for VFQ-25 score outcomes. The VFQ-25 score is significantly correlated with RA and the amount of screen time. Points show the VFQ-25 score for each participant assessed. The amount of screen time of participants is color-coded. Abbreviations: VFQ, visual functioning questionnaire.

**Table 1 jpm-13-00463-t001:** Baseline characteristics of participants in this study.

Characteristic	No. (%)
	Healthy Group (*n* = 207)
Age, mean (SD), y	22.46 (4.01)
Gender	
Male	116 (56.04)
Female	91 (43.96)
Education	
Undergraduate	146 (71.22)
Postgraduate	26 (12.68)
Doctor and Postdoc	33 (16.10)
Myopia, eyes	
No	40 (9.66)
Low	144 (34.78)
Moderate	187 (45.17)
High	43 (10.39)
Screen time, mean (SD), h	
Phone	4.55 (1.37)
Personal computer	4.32 (1.65)
Pad	2.55 (1.71)
Total	5.06 (1.50)
VFQ-25, mean (SD)	86.26 (9.11)

Abbreviations: VFQ, visual functioning questionnaire.

**Table 2 jpm-13-00463-t002:** Reading characteristics of participants in this study.

Characteristic	Mean (SD)
	Healthy Group (*n* = 191)
RA, logMAR	
both eyes	0.193 (0.104)
right eye	0.242 (0.124)
left eye	0.249 (0.120)
MRS, wpm	
both eyes	185.16 (44.93)
right eye	171.65 (46.27)
left eye	168.59 (45.68)
CPS, logMAR	
both eyes	0.419 (1.05)
right eye	0.412 (0.647)
left eye	0.371 (0.229)

Abbreviations: RA, reading acuity; MAR, minimum angle of resolution; MRS, maximum reading speed; wpm, words per minute; CPS, critical print size.

**Table 3 jpm-13-00463-t003:** Left- and right-eye reading characteristics for different genders, education levels, and myopia levels.

		RA	MRS	CPS
Gender	Male (*n* = 105)	0.17 (0.10)	184.95 (52.74)	0.27 (0.13)
	Female (*n* = 86)	0.22 (0.11)	185.42 (33.29)	0.34 (0.15)
	*p* value	0.005	0.685	0.025
Education	Undergraduate (*n* = 133)	0.19 (0.11)	178.57 (48.40)	0.30 (0.15)
	Postgraduate (*n* = 26)	0.17 (0.06)	203.76 (39.07)	0.27 (0.09)
	Doctor and Postdoc (*n* = 30)	0.21 (0.10)	199.64 (21.80)	0.33 (0.14)
	*p* value	0.217	0.002	0.093
Myopia	No (*n* = 37)	0.16 (0.08)	162.4 (42.9)	0.29 (0.10)
	Low (*n* = 136)	0.19 (0.11)	163.2 (46.1)	0.34 (0.15)
	Moderate (*n* = 166)	0.19 (0.11)	178.6 (43.4)	0.34 (0.14)
	High (*n* = 43)	0.22 (0.08)	162.9 (44.7)	0.44 (0.18)
	*p* value	0.772	0.049	< 0.001

Abbreviations: RA, reading acuity; MRS, maximum reading speed; CPS, critical print size.

**Table 4 jpm-13-00463-t004:** Parameter estimates for multivariate linear regression model of RA, MRS, and CPS in a single eye.

Measure		Multivariate Linear Regression		Stepwise Regression (*p* value = 0.05)	
		β (95% CI)	*p* Value	β (95% CI)	*p* Value
RA	Intercept	0.213 (0.128 to 0.297)	<0.001	0.22 (0.19 to 0.24)	<0.001
	Age, y	−0.00074 (−0.00387 to 0.00239)	0.642		
	Screen time, h	0.0044 (−0.0045 to 0.0133)	0.331		
	Myopia, D	−0.012 (−0.018 to −0.006)	<0.001	−0.012 (−0.018 to −0.006)	<0.001
MRS	Intercept	179.77 (152.39 to 207.15)	<0.001	166.0 (150.9 to 181.1)	<0.001
	Age, y	−0.55 (−1.56 to 0.47)	0.289		
	Screen time, h	3.45 (0.57 to 6.33)	0.019	3.19 (0.34 to 6.04)	0.028
	Myopia, D	0.68 (−1.33 to 2.69)	0.507		
CPS	Intercept	0.19 (0.09 to 0.30)	<0.001	0.23 (0.16 to 0.29)	<0.001
	Age, y	−0.0014 (−0.024 to 0.005)	0.467		
	Screen time, h	0.015 (0.004 to 0.026)	0.010	0.015 (0.004 to 0.026)	0.007
	Myopia, D	−0.013 (−0.021 to −0.006)	<0.001	−0.013 (−0.021 to −0.006)	<0.001

Abbreviations: D, diopter; RA, reading acuity; MRS, maximum reading speed; CPS, critical print size.

**Table 5 jpm-13-00463-t005:** Parameter estimates for multivariate linear regression and stepwise regression model for the outcomes VFQ-25 score.

Measure		Multivariate Linear Regression		Stepwise Regression (*p* Value = 0.05)	
		β (95% CI)	*p* Value	β (95% CI)	*p* Value
Right eye	Intercept	109.76 (99.15 to 120.38)	<0.001	100.82 (94.78 to 106.86)	<0.001
	Age, y	−0.17 (−0.50 to 0.15)	0.298		
	Total screen time, h	−0.86 (−1.30 to −0.42)	<0.001	−0.86 (−1.29 to −0.42)	<0.001
	Refraction, D	0.42 (−0.17 to 1.01)	0.164		
	RA, logMAR	−16.60 (−32.19 to −1.00)	0.037	−18.65 (−30.88 to −6.43)	0.003
	MRS, wpm	−0.017 (−0.046 to 0.012)	0.240		
	CPS, logMAR	−2.75 (−14.36 to 8.86)	0.640		
Left eye	Intercept	105.58 (95.57 to 115.60)	<0.001	100.68 (95.15 to 106.21)	<0.001
	Age, y	−0.08 (−0.37 to 0.21)	0.598		
	Total screen time, h	−0.64 (−1.05 to −0.23)	0.002	−0.63 (−1.04 to −0.23)	0.002
	Refraction, D	0.11 (−0.43 to 0.65)	0.683		
	RA, logMAR	−21.31 (−35.69 to −6.92)	0.004	−26.83 (−37.84 to −15.83)	<0.001
	MRS, wpm	−0.01 (−0.04 to 0.02)	0.429		
	CPS, logMAR	−6.28 (−17.37 to 4.82)	0.266		
Both eyes *	Intercept	107.25 (96.62 to 117.89)	<0.001	100.24 (94.59 to 105.89)	<0.001
	Age, y	−0.12 (−0.44 to 0.20)	0.451		
	Total screen time, h	−0.88 (−1.32 to −0.44)	<0.001	−0.86 (−1.29 to −0.43)	<0.001
	RA, logMAR	−15.61 (−30.70 to −0.52)	0.043	−21.41 (−33.74 to −9.08)	0.001
	MRS, wpm	−0.015 (−0.045 to 0.015)	0.330		
	CPS, logMAR	−7.59 (−18.30 to 3.12)	0.164		

Abbreviations: D, diopter; RA, reading acuity; MAR, minimum angle of resolution; MRS, maximum reading speed; wpm, words per minute; CPS, critical print size; VFQ, visual functioning questionnaire. *: Refraction was not included in the regression model for both eyes since refraction was not significant for VFQ-25 scores in both the left and right eyes.

## Data Availability

Data will be made available on request.

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
