# Peer review of "Using the C-Read as a Portable Device to Evaluate Reading Ability in Young Chinese Adults: An Observational Study"

_jpm, 2023, doi:10.3390/jpm13030463_

Round 1

Reviewer 1 Report

Title:

Please, indicate the study’s design with a commonly used term (e.g., observational study) in the title and/or in the abstract

Introduction:

In my opinion, the relationship between myopia and reduced reading acuity/speed should be better described. 

To justify the need for this study, the authors should describe whether any other research has studied the relationship between quality of life and reading acuity/speed. In addition, other studies showing normal values for RA, CPS and MRS should be mentioned in the introduction

The title of the study refers to a "pilot trial", but the objectives relate to an epidemiological study (values in a healthy population) and to a study that analyses the relationship between reading speed and quality of life. So, is it a pilot study? If so, the objectives should be characteristics of a pilot study, like determining the minimum sample size for a broader study and/or exploring some possible associations.

Methods:

Please describe the methods of selection of participants, how did you verify that the participants were "healthy"? and how do you define "healthy"?

How did the authors verify that participants had "no strabismus, amblyopia, or any other ocular or systemic diseases that may affect reading ability"?

lines 123-124: rather than "well-lit", the authors should state the light level in lux

lines 136-137: Sorry, but I don't understand the sentence "the C-Read application on the smartphone can be used to switch 136 measurements when checking different eyes"

Please indicate the units used for measuring the main variables (RA, MRS and CPS). If possible, indicate these variables' normal values according to previous studies 

In my opinion, the authors should describe any efforts implemented to address potential sources of bias (e.g., selection bias)

Was the refractive status self-declared by participants?

What was the criteria for determining each participant's myopia level?

Please explain how the study size was arrived at

In order to ensure sample representativeness, did the authors employ any sampling strategy?

Describe all statistical methods

For the analyses, did the authors use both eyes or one eye? Justify this decission

Results:

The authors used parametric statistical tests but, did the variables show normal distribution?

lines 181-185: Please check the units for measuring each variable

Figure 4 and Figure 5 would be easier to interpret if they show the regression line

C-test values for healthy populations are not provided

Discussion:

Please discuss how the sample characteristics may influence the generalisability (external validity) of the study results

Author Response

Response to Reviewer 1 Comments

Point 1: Please, indicate the study’s design with a commonly used term (e.g., observational study) in the title and/or in the abstract.

Response 1: Thank you for your kindly reminder. Article title has been changed to " Using the C-Read to measure reading ability in young Chinese adults: an observational study".

Point 2: In my opinion, the relationship between myopia and reduced reading acuity/speed should be better described.

Response 2: The relationship between myopia and reading ability is complex and not a single linear relationship. Reading ability is affected by myopia, but also by the difficulty of the reading material, light, etc. Due to the characteristics of the Chinese script, there is a difference between reading Chinese in general and reading the sight chart. Based on the literature review, few studies concentrated on the relationship between the two variances. Therefore, we would like to look at the relationship between the two through this study.

Point 3: To justify the need for this study, the authors should describe whether any other research has studied the relationship between quality of life and reading acuity/speed. In addition, other studies showing normal values for RA, CPS and MRS should be mentioned in the introduction.

Response 3: Little is known about the reading characteristics of young people with normal vision whose native language is Chinese, due to the lack of a reliable Chinese reading chart to assess Chinese reading ability. Han et al. measured the reading characteristics of 118 Chinese college students between the ages of 18-24 who had normal or corrected vision, and obtained normal mean (SD) reference values for RA, CPS and MRS of 0.16(0.05) logMAR, 0.24(0.06) logMAR, and 273.44(34.37) Chinese wpm, respectively. In addition, previous studies have observed a clear correlation between reading speed and vision-related QoL in people with eye disease, and the results of a study by an Italian academic point to both reading speed and visual acuity as important determinants of quality of life. However, it remains to be seen whether this correlation holds in the healthy population.

Point 4: The title of the study refers to a "pilot trial", but the objectives relate to an epidemiological study (values in a healthy population) and to a study that analyses the relationship between reading speed and quality of life. So, is it a pilot study? If so, the objectives should be characteristics of a pilot study, like determining the minimum sample size for a broader study and/or exploring some possible associations.

Response 4: Thank you for your kindly reminder. One of the objectives of this study was to evaluate the reading characteristics of normally sighted young adults over a wider age range (18-35 years) using a C-Read test to provide baseline values in a healthy population, so this was really a pilot study. However, it is true that the characteristics of the pilot study are not reflected in the article, so the title of this article has been revised.

Point 5: Please describe the methods of selection of participants, how did you verify that the participants were "healthy"? and how do you define "healthy"?

Response 5: All subjects' eye health status was confirmed by a professional ophthalmologist. Healthy participants were included according to the following criteria: (1) age between 18 and 35 years, (2) good physiological functioning of the body's systems without health problems requiring medical intervention, (3) no strabismus, amblyopia, or any other ocular or systemic diseases that may affect reading ability, (4) good cooperation in the questionnaire survey and reading tests

Point 6: How did the authors verify that participants had "no strabismus, amblyopia, or any other ocular or systemic diseases that may affect reading ability"?

Response 6: The professional ophthalmologist examined the subjects for distance and near vision, slit lamp observation, and indirect ophthalmoscopy.

Point 7: lines 123-124: rather than "well-lit", the authors should state the light level in lux

Response 7: The mean luminance on the device was 100 cd/m2.

Point 8: lines 136-137: Sorry, but I don't understand the sentence "the C-Read application on the smartphone can be used to switch 136 measurements when checking different eyes"

Response 8: First, 136 is the line number of the article, there is no 136 in this sentence. This passage has been revised to read "Three scales, A, B, C, were used to assess participants' right, left and binocular reading ability. When changing eyes, the examiner can switch to a different scale for the next test using the C-Read application on the smartphone".

Point 9: Please indicate the units used for measuring the main variables (RA, MRS and CPS). If possible, indicate these variables' normal values according to previous studies

Response 9: Thank you for your kindly reminder. The units used to measure RA, MRS and CPS are logMAR, wps and logMAR respectively. Little is known about the reading characteristics of young people with normal vision whose native language is Chinese. Han et al. obtained normal mean (SD) reference values for RA, CPS and MRS of 0.16(0.05) logMAR, 0.24(0.06) logMAR, and 273.44(34.37) Chinese wpm, respectively.

Point 10: In my opinion, the authors should describe any efforts implemented to address potential sources of bias (e.g., selection bias)

Response 10: This study used a random sampling method to reduce potential bias.

Point 11: Was the refractive status self-declared by participants?

Response 11: The subjects were examined for corrected visual acuity by professional ophthalmologists. All subjects met a corrected visual acuity of 1.0 and no definite eye disease was detected. Meanwhile, where the subjects satisfied a corrected visual acuity of 1.0, the participants' refractive status was self-declared.

Point 12: What was the criteria for determining each participant's myopia level?

Response 12: Thank you for your kindly reminder. Myopia between −0.00 and −0.50 diopters is classified as emmetropia. Low, moderate, and high myopia describe myopia between −0.50 and −3.00, -3.00 and -6.00, −6.00 or more diopters, respectively.

Point 13: Please explain how the study size was arrived at

Response 13: Thank you for your kindly reminder. Set σ = 7δ, α = 0.05, run a two-sided test and work out that the sample needs 189 people. Considering a 5% missing rate, a total of at least 200 people need to be recruited at first.

Point 14: In order to ensure sample representativeness, did the authors employ any sampling strategy?

Response 14: Participants were recruited across different academic departments at Peking University (including Humanities, Science, Social Sciences and Medicine). Within each academic department, we use simple random sampling to obtain the sample.

Point 15: Describe all statistical methods

Response 15: Thank you for your kindly reminder. After the revision, all statistical methods have been clearly described.

Point 16: For the analyses, did the authors use both eyes or one eye? Justify this decision.

Response 16: Our study was conducted using three scales from the C-ReadF device, testing the left and right eyes of both eyes respectively. In order to avoid any omissions, both eyes, the left eye and the right eye were tested separately. Previous studies by Han et al. have also used data from both eyes for their analyses. In contrast our research is more comprehensive and rigorous.

Point 17: The authors used parametric statistical tests but, did the variables show normal distribution?

Response 17: Thank you for your kindly reminder. All variables using the parametric tests obeyed a normal distribution. This was tested by P-P plots, which will be added to supplementary files.

Point 18: lines 181-185: Please check the units for measuring each variable

Response 18: Thank you for your kindly reminder. The error in the unit here has been corrected.

Point 19: Figure 4 and Figure 5 would be easier to interpret if they show the regression line

Response 19: We have not chosen to add regression lines to Figures 4 and 5 since both of these are bubble plots. Bubble plots are suitable for multivariate linear regressions and drawing further lines when representing a multivariate linear regression would be misleading. If a line is drawn, it will only reflect a linear regression of a single variable.

Point 20: C-test values for healthy populations are not provided

Response 20: Does C-test stand for Clinical Tests? This article only considers data from the normal population; data from the patient population and the diagnostic thresholds are for future studies.

Point 21: Please discuss how the sample characteristics may influence the generalizability (external validity) of the study results

Response 21: We would like to do a standardized, random sample study of a younger population and hope that the results of this study will provide the basis for our follow-up study. In our follow-up study, we will compare children and older people, as well as patients with eye disease. We have mentioned sample selection bias in our limitation part.

Reviewer 2 Report

Cheng and coauthors evaluated reading speed using the C-Read in normal-sighted young adults. The C-Read is a smartphone-based portable device for testing reading speed. This pilot study provided a baseline value for the C-Read in normal-sighted young adults. Their results revealed that myopia was a significant factor affecting reading acuity and CPS, while screen-use time significantly affected MRS. Moreover, the reading acuity and PCS are significantly different between sex and increase with the progression of myopia. I have some comments and questions that may help to improve this manuscript as follows:

- How is the C-Read different from other reading test systems?

- What does the C-Read system able to measure reading equity and speed in Chinese?

- If the C-Read is designed for Chinese, how can it be used for English or other linear alphabetic languages? Or the C-Read is developed only for the Chinese?

- It would be great if the authors provide pictures or videos of the actual test screen as supplementary files to help readers understand.

- Please double-check the format of Table 1, column 1. It was not well organized and hard to read.

- Please check carefully for errors throughout the manuscript. There are some errors. (e.g. two periods in line 273, VFO-25 score in line 354)

Author Response

Response to Reviewer 2 Comments

Point 1: How is the C-Read different from other reading test systems?

Response 1: The C-Read is a set of three visual ability scales for assessing reading competency in Chinese. Each scale consists of 16 Chinese sentences, each consisting of 12 simplified Chinese characters, selected, and adapted from textbooks for grades 1 to 3. The length and content of the sentences are carefully selected according to the characteristics of the Chinese language (e.g. sentences with subordinate clauses are not selected, and each sentence contains the same number of simple or complex characters), ensuring that the test tool is adapted to the specific needs of Chinese reading tests. The C-Read system includes a high-definition screen with intelligent display and the ability to synchronize with smartphone application via Bluetooth. The device has a voice recording function and is able to store test data and record patient details.

Point 2: What does the C-Read system able to measure reading equity and speed in Chinese?

Response 2: Reading speed data in log character per minute collected from a C-READ test is plotted against the chart print size in logMAR. C-Read uses this chart to measure reading equity and speed using the same method as MNRead. The height of the horizontal line segment is the maximum reading speed. The horizontal position of the intersection point between the sloped and horizontal line segments is the critical print size. Floor effect is frequently observed in the C-Read test because of the presence of simple and readable characters in Chinese sentences. C-Read program automatically removes all characters in the floor effect except for the largest character in size, producing a monotonically rising linear part of the reading speed function used to calculate the C-Read parameter .

Point 3: If the C-Read is designed for Chinese, how can it be used for English or other linear alphabetic languages? Or the C-Read is developed only for the Chinese?

Response 3: C-Read is a reading test developed specifically for the Chinese population and is inspired by many of the design principles embodied in many of the English reading visuals such as MNRead and the Radner.

Point 4: It would be great if the authors provide pictures or videos of the actual test screen as supplementary files to help readers understand.

Response 4: Thank you for your kindly reminder. Pictures of the actual test screen has been added to supplementary files.

Point 5: Please double-check the format of Table 1, column 1. It was not well organized and hard to read.

Response 5: Thank you for your kindly reminder. Table 1 has been reformatted.

Point 6: Please check carefully for errors throughout the manuscript. There are some errors. (e.g. two periods in line 273, VFO-25 score in line 354)

Response 6: Thank you for your kindly reminder. These errors have been corrected.

Round 2

Reviewer 1 Report

1. Introduction:

1.1. (lines 52-55): In my opinion, for a better understanding of the usefulness of this research, the authors should outline a description of the relationship between myopia and reading. The sentence "many patients with refractive error 53 and vision impairment have reading difficulty that directly affects the quality of life 54 (QoL)" needs a reference and it must be specified that vision impairment is not the same as myopia, despite myopia being a risk factor for vision impairment. Moreover, the authors should justify why the studied the suggested relationship in myopia and not in other refractive error

1.2. (lines 115-117): The authors should clarify why it is pertinent to study reading characteristics in young adults not included in their previous study (24-35 years)

2. Materials and methods:

2.1. (lines 140-144): For a better interpretation of the results, it must be mentioned that level of myopia was self-declared by participants and if the level of myopia indicated described right eye, left eye of both. As level of myopia is indeed related with binocular vision status and the latter has been shown to be related with reading problems, it should be clarified whether binocular vision and ocular accomodation was cheched in anyway

2.2. (lines 155-157): The questionnaire administered along with the VFQ-25 must be provided along with the manuscript

2.3. (line 166): Sample image included in Figure 2 and Figure S1 dos not shoy any headrest

2.4. Please explain how study size was arrived at

2.5. A table, provided as supplemetary material, showing meand +- SD results for VFQ25 sub-scales will be useful for interpreting the results

2.6. Were the differences between monocular and binocular reading performance values significant?

2.7. (lines 263-270 and Figure 3). Were these differences found in monocular, binocular or in both reading performance values? 

2.8. Please justify the use of stepwise regression instead of other methods

2.9. Please state whether you used monocular or binocular data for the regression analysis and justify your election

3. Discussion

3.1. Please discuss differences by sex group, it is just mentioned in the discussion

3.2. Please discuss differences between monocular and binocular measures

3.3. Please discuss limitaions related to the use o self-reported myopia status

4. Conclusions

It should be reviewed once the authors provide the previously requested information

Reviewer 2 Report

Comments from the previous review are well addressed by the authors in the revised manuscript. 
